# Biosynthesis of Fatty Acid Derivatives by Recombinant *Yarrowia lipolytica* Containing MsexD2 and MsexD3 Desaturase Genes from *Manduca sexta*

**DOI:** 10.3390/jof9010114

**Published:** 2023-01-14

**Authors:** Jaroslav Hambalko, Peter Gajdoš, Jean-Marc Nicaud, Rodrigo Ledesma-Amaro, Michal Tupec, Iva Pichová, Milan Čertík

**Affiliations:** 1Institute of Biotechnology, Faculty of Chemical and Food Technology, Slovak University of Technology, Radlinského 9, 812 37 Bratislava, Slovakia; 2Université Paris-Saclay, INRAE, AgroParisTech, Micalis Institute, Domaine de Vilvert, 78352 Jouy-en-Josas, France; 3Department of Bioengineering and Imperial College Centre for Synthetic Biology, Faculty of Engineering, Imperial College, London SW7 2AZ, UK; 4Institute of Organic Chemistry and Biochemistry of the Czech Academy of Sciences, Flemingovo náměstí 542/2, 160 00 Prague 6, Czech Republic

**Keywords:** *Yarrowia*, pheromone, desaturase, *Manduca*, conjugase, metabolic engineering

## Abstract

One of the most interesting groups of fatty acid derivates is the group of conjugated fatty acids from which the most researched include: conjugated linoleic acid (CLA) and conjugated linolenic acid (CLNA), which are associated with countless health benefits. Sex pheromone mixtures of some insect species, including tobacco horn-worm (*Manduca sexta*), are typical for the production of uncommon C16 long conjugated fatty acids with two and three conjugated double bonds, as opposed to C18 long CLA and CLNA. In this study, *M. sexta* desaturases MsexD2 and MsexD3 were expressed in multiple strains of *Y. lipolytica* with different genotypes. Experiments with the supplementation of fatty acid methyl esters into the medium resulted in the production of novel fatty acids. Using GCxGC-MS, 20 new fatty acids with two or three double bonds were identified. Fatty acids with conjugated or isolated double bonds, or a combination of both, were produced in trace amounts. The results of this study prove that *Y. lipolytica* is capable of synthesizing C16-conjugated fatty acids. Further genetic optimization of the *Y. lipolytica* genome and optimization of the fermentation process could lead to increased production of novel fatty acid derivatives with biotechnologically interesting properties.

## 1. Introduction

In recent years, there has been an increasing demand for attractive lipids by the chemical industry. Lipids in the form of fats and oils as renewable sources are environmentally interesting substances with many industrial applications. In the industry, lipids can be used as precursors in the synthesis of polymers, lubricants, plasticizers, surfactants, coatings, drugs, fuels, and others [1]. In nature, lipids are stored in the animal, plant, and microbial cells, primarily in the form of triacylglycerols (TAG). In contrast to animals and plants, microorganisms are genetically engineered more easily, and genetically modified microorganisms are more easily accepted by industry and society [2]. The recent metabolic engineering development facilitates the direct production of unusual fatty acids by modified microorganisms [3,4]. The most promising microbial oil producers belong to the group called oleaginous microorganisms, defined by the ability to accumulate more than 20% of their cell dry weight as lipids.

One of the microorganisms that could make a perfect cell factory for the industrial production of fatty acid derivatives is *Yarrowia lipolytica* [5]. *Y. lipolytica* is an oleaginous, dimorphic, and non-pathogenic yeast that exhibits remarkable lipolytic and proteolytic activities. The genome of *Y. lipolytica* has been known for a long time. Thanks to our knowledge of this yeast, the development of the tools for manipulating the genome of *Y. lipolytica* makes this strain a textbook organism for the biosynthesis of unusual fatty acids study [2].

Recent years have witnessed continuous growth in the interest in the production of conjugated fatty acids (FAs), such as conjugated linoleic acid (CLA) and conjugated linolenic acid (CLNA), which are associated with countless health benefits. CLA is the only group of conjugated FAs whose production was described in *Y. lipolytica* [6,7,8].

Recently, increased attention has been paid to the study and identification of enzymes that catalyze the formation of conjugated double bonds. These enzymes are called conjugases. Some desaturases exhibit both desaturase and conjugase activity, and these enzymes are able to produce their substrates as well [9,10]. Some insect species, including the tobacco hornworm (*Manduca sexta*), are typical for the production of uncommon C16 long conjugated FA with two (2UFA) and three conjugated double bonds (3UFA), as opposed to C18 long CLA and CLNA.

Derivates of these C16 2UFA and 3UFA are the main components of the pheromone blend mixture produced by the tobacco hornworm (*Manduca sexta*) females, which is a pest belonging to the order Lepidoptera. Buček et al. 2015, identified 14 desaturase transcripts in *M. sexta*, of which 4 were abundant and enriched in the pheromone gland. One of those desaturases was previously characterized by bi-functional MsexD2 (Δ11 desaturase with conjugase activity) involved in C16:1Δ^cis11^ mono- and C16:2Δ^trans10,trans12^ and C16:2Δ^trans10,cis12^ diunsaturated fatty acids (FA) biosynthesis. Three newly identified desaturases were MsexD3, MsexD5, and MsexD6. The MsexD3 desaturase catalyzes the biosynthesis of C16:3Δ^trans10,trans12,cis14^ and C16:3Δ^trans10,trans12,trans14^ triunsaturated fatty acids from diunsaturated FA via Δ14 desaturation. Specificities of both MsexD2 and MsexD3 are influenced by a character of amino acids forming the binding tunnel for fatty acid substrates [11,12].

In this study, multiple strains of *Y. lipolytica* with different genotypes were constructed for the expression of MsexD2 and MsexD3 FADs from *M. sexta* and the production of biotechnologically valuable long-chain conjugated fatty acids.

## 2. Materials and Methods

### 2.1. Strains, Media Composition, and Culture Conditions

All the *Y. lipolytica* and *Escherichia coli* strains used in this study are listed in Table 1. The *E. coli* strains were cultured in an LB (lysogeny broth) medium containing a required antibiotic (50 mg/mL of kanamycin or 100 mg/mL of ampicillin) [13]. Strain W29 (ATCC 20460) was used as the strain from which all other recombinant strains have been derived. *Y. lipolytica* transformants were selected on minimal YNB, YNBUra, and YNBLeu media agar plates. The minimal YNB medium contained 1.7 g/L of yeast nitrogen base (without amino acids and ammonium sulfate; BD, Erembodegem, Aalst, Belgium), 5 g/L of ammonium chloride, 50 mM of phosphate buffer with pH 6.8, and 20 g/L of glucose. The YNBUra and YNBLeu media contained 0.1 g/L of uracil and leucine, respectively, as an addition to the YNB medium. Agar at a concentration of 20 g/L was added to the YNB media to prepare solid agar plates. For *Y. lipolytica*, an inoculum consisting of a rich YPD medium containing 10 g/L of yeast extract (BD, Erembodegem, Aalst, Belgium), 10 g/L of peptone (BD, Erembodegem, Aalst, Belgium), and 20 g/L of glucose (Mikrochem, Pezinok, Slovakia was used). The yeast inoculum was prepared in 20 mL of the YPD medium in 100 mL flasks, and 24 h old inoculum with an optical density (OD600) of 0.1 was used for the inoculation of production media. For the lipid production MedA^+^ medium composed of 1.5 g/L yeast extract, 0.5 g/L NH_4_Cl, 7 g/L KH_2_PO_4_, 5 g/L Na_2_HPO_4_.12H_2_O, 0.1 g/L CaCl_2_, 1.5 g/L MgSO_4_.7H_2_O, 10 mg/L ZnSO_4_.7H_2_O, 0.6 mg/L FeCl_3_.6H_2_O, 0.07 mg/L MnSO_4_.H_2_O, and 0.04 mg/L CuSO_4_.5H_2_O was used. The carbon source used in MedA^+^ media was either glucose or crude glycerol (Mikrochem, Pezinok, Slovakia) in a concentration of 60 g/L. The high C/N ratio (C/N = 80) of this medium makes it suitable for the accumulation of lipids in yeasts. The MedA^+^ growth medium was prepared by the modification of the MedA medium [14]. Fifty mL of inoculated production medium in 250 mL baffled flasks were cultured at 28 °C and 130 rpm inside an orbital shaker (Innova 40R, Hamburg, Germany). In order to produce the desired fatty acids, the strains of *Y. lipolytica* were cultured for 3 days. The selected strain was co-cultivated with fatty acid methyl esters (FAME) as precursors for 3 days. FAMEs were dissolved in ethanol and added directly to the medium before cultivation to a final concentration of 0.25 mM. All the experiments were performed in three independent biological replicates.

### 2.2. Plasmid and Strain Construction

The *MsexD2* and the *MsexD3* genes were codon-optimized for *Y. lipolytica* (Appendix A), synthesized, and the fragments were digested using BamHI and AvrII endonucleases. The fragments treated in this manner were then inserted into corresponding BamHI and AvrII sites on the already-available plasmids JME1046 [15] and JME2563 containing *pTEF* promoter and JME2607 and JME3048, which contained the *8UAS-pTEF* promoter [16]. All the vectors were based on the JMP62 vector with URA3 and LEU2 selectable markers to complement the LEU and URA auxotrophy, respectively, in the final strain. Before being used to transform *Y. lipolytica* with the lithium acetate method [19], the plasmids were digested with NotI. The transformants were selected on the appropriate medium, and subsequently, the genomic DNA was isolated from the yeast transformants [20]. In order to confirm positive transformants, the PCR amplifications were performed in an Eppendorf 2720 thermal cycler using GoTaq DNA polymerases (Promega). The PCR fragments were purified with QIAgen Purification Kit (Qiagen, Hilden, Germany) and verified by gel electrophoresis and sequencing. The manufacturer’s instructions were followed in all performed reactions.

All strains prepared for this work were derived from the wild-type strain W29. The first host strain for insertion of the *FAD* gene was Po1d, which was constructed directly from the W29. Along with the FAD coding genes, yeast was transformed with the *Leu2* gene resulting in two new prototrophic strains. JMY6699 expresses the *MsexD2* gene, and JMY6700 expresses the *MsexD3* gene, both under the control of the *pTEF* promoter. As the second host for the expression of FADs, the selected strain JMY3820 was prepared from the prototrophic JMY3501. Both JMY3501 and JMY3820 were prepared from the JMY1233 [21] strain in a study published by Lazar et al. 2014. In total, three strains expressing FADs under the control of *8UAS-pTEF* promoter were constructed: JMY7078 (*MsexD2*), JMY7080 (*MsexD3*), and JMY7084 (*MsexD2* and *MsexD3*).

### 2.3. Analytical Methods

Yeast cells were centrifuged (2880× *g*, 5 min) and separated from the media. The pellet of cells was washed twice with the saline solution (NaCl, 9 g/L) and once with deionized water, and then suspended in deionized water and lyophilized. Lyophilisates were used for the determination of dry cell weight (DCW) and the analysis of lipids.

The residual amount of carbon substrates in media was determined with HPLC (Agilent Technologies, Santa Clara, CA, USA) using Aminex HPX87H column (Bio-Rad, Hercules, CA, USA) coupled with RI and DAD detectors. The flow rate of the mobile phase (H_2_SO_4_, 5 mM) was 0.6 mL/min [22].

In order to directly prepare methyl esters of fatty acids from the biomass, the freeze-dried cells (approximately 10 mg) were added to a mixture of 1 mL CH_2_Cl_2_ (containing 0.1 mg of C13:0 as the internal standard) and 2 mL anhydrous methanolic HCl solution. The suspension was then incubated at 50 °C for 3 h. Subsequently, 1 mL of each water and hexane were added to the mixture with the sample, and the whole suspension was vortexed and centrifuged (2880× *g*, 5 min). The organic layer FAME was collected and analyzed using GC-6890 N (Agilent Technologies, Santa Clara, CA, USA). The samples (1 μL) were injected automatically into the DB-23 column (50% cyanopropylmethylpolysiloxane, length 60 m, diameter 0.25 mm, film thickness 0.25 mm) and analyzed. The analysis conditions were: carrier gas–hydrogen, inlet (230_C; hydrogen flow: 37 mL/min; split–10:1), FID detector (250 °C, hydrogen flow: 40 mL/min, air flow: 450 mL/min.), gradient (150 °C–0 min; 150–170 °C–5.0 °C/min; 170–220 °C–6.0 °C/min; 220 °C–6 min; 220–230 °C–6 °C/min; 230 °C–1 min; 230–240 °C–30 °C/min; 240 °C–6 min). The chromatograms were analyzed using the Agilent Open LAB CDS C.01.07 SR7 software. The amounts of fatty acids were quantified according to the peak area normalized using C13:0 as the internal standard. The fatty acids were identified according to the C4–C24 FAME standard (Supelco, Bellefonte, PA, USA). To confirm the identity of the obtained peaks, the GC-MS was performed (EI at 70 eV) according to their MS spectra.

Analysis of fatty acid methyl esters was also performed using a 6890N gas chromatograph (Agilent Technologies) coupled to Pegasus IV D time-of-flight mass selective detector (LECO Corp., St. Joseph, MI, USA). The sample (1 μL) was injected through an inlet (250 °C; split—10:1) onto the primary column Rxi-5Sil MS (length 32 m, diameter 0.25 mm, film thickness 0.25 μm) connected to the secondary column Rxi-17Sil MS (length 1.9 m, diameter 0.15 mm, film thickness 0.15 μm). The separation conditions were as follows: carrier gas–helium (1 mL/min), temperature gradient (100 °C–1 min; 100 °C→285 °C–4 °C/min; 285 °C→320 °C–20 °C/min; 320 °C–2 min), secondary oven temperature offset (relative to primary oven): +10 °C, modulator temperature offset (relative to secondary oven): +20 °C, modulation time: 4 s (hot pulse time 0.8 s, cool time 1.2 s). The MS detector was operated in electron ionization mode (transfer line temperature: 260 °C, ion source temperature: 220 °C, electron voltage: −70 V, detector voltage: 1500 V). The chromatograms were analyzed using the LECO ChromaTOF 4.72 software.

For analysis of the proportion of individual lipid structures and composition of FA in lipid structures, total cellular lipids were extracted using chloroform: methanol (2:1) solution [23]. For the analysis of the proportion of lipid structures, the organic extracts were loaded by CAMAG ATS 4 automatic sampler on Merck thin-layer chromatography (TLC) silica gel 60 plates and developed in a closed cuvette filled with a hexane/ether/acetic acid (70:30:1) system. The developed plates were briefly immersed in the aqueous solution of 50% (*v*/*v*) methanol, 3.3% (*v*/*v*) sulphuric acid, and 0.33% (*w*/*v*) MnCl2. Afterward, they were dried at 130 °C for 10 min for visualization. The plates were then scanned at 400 nm by the CAMAG TLC Scanner 4 and evaluated using the software winCATS ver. 1.4.8 (CAMAG) [24]. For the analysis of the composition of FA in lipid structures, the lipids were separated by TLC, as described previously, and visualized using iodine vapors. The identified separated lipid bands were scraped off into test tubes. Subsequently, the FA was transesterified [25], and the FA methyl esters were analyzed by gas chromatography [24].

## 3. Results

### 3.1. Growth and Production of Fatty Acid Derivatives in Y. lipolytica Strains Expressing MsexD2 and MsexD3 FADs

Sequences of *MsexD2* and *MsexD3* genes were codon optimized for *Y. lipolytica*, synthesized, cloned into JMP62 vector backbone, and transformed into *Y. lipolytica* Po1d strain which is *leu2-* and *ura3-*auxotroph prepared from the wild type strain W29 [17]. The resulting strains carrying MsexD2 and MsexD3 sequences were termed JMY6699 and JMY6700, respectively.

*Y. lipolytica* was grown on two different media with glucose as a carbon source, YPD, and MedA^+^. YPD medium simulates non-oleaginous conditions, while MedA^+^ medium strongly favors the formation of lipids. YPD is a very rich medium on which yeasts are growing faster, but they are not accumulating higher amounts of storage lipids. To compare strains in different conditions at approximately the same stage of growth, yeasts on the YPD medium were grown for 48 h and 72 h on the MedA+ medium. W29 was used as a control strain for this cultivation.

Figure 1 indicates that the biomass yield and accumulation of lipids were similar in all strains, except for an obvious difference caused by using two different media. Lipid accumulation was lower in YPD medium, which is consistent with the assumption that the YPD medium does not have a high C/N ratio and is not suitable for lipid overproduction.

The most abundant fatty acid produced by both desaturases was C16:1Δ^11^. While JMY6700 (MsexD3) produced trace amounts of C16:1Δ^11^ only under oleaginous conditions (in MedA^+^ medium with C/N = 80), JMY6699 (*Msex*D2) produced C16:1Δ^11^ (9.6% of total fatty acids; TFA), C17:1Δ^11^ (trace amount), and C18:1Δ^11^ (2% of TFA) in both media (Figure 2). These results demonstrate the Δ11-desaturase activity of *Msex*D2 and *Msex*D3 in *Y. lipolytica*. However, the production of FA with two (2UFAs) and three (3UFAs) conjugated double bonds was not detected. Figure 3 illustrates that both C16:1Δ^11^ and C18:1Δ^11^ were accumulated in all lipid classes in the cells of JMY6699. Most of the new FAs were found among polar lipids. However, it is common for *Y. lipolytica* to incorporate a higher level of unsaturated FA than saturated into the polar lipids, which form the membrane structures. The results might suggest that new FAs were not toxic to the cells. Since the number of new metabolites was too low, the engineering of *Y. lipolitica* strains genetic was required.

### 3.2. Metabolic Engineering of Y. lipolytica for Effective Production of FA Derivatives

To optimize FA production by *M. sexta* FADs in *Y. lipolytica,* the expression of FADs was driven by a stronger *8UAS-pTEF* promoter in *Y. lipolytica* JMY3820. JMY3820 was engineered for the accumulation of high amounts of storage lipids. It has deleted all the six *POX* genes and *TGL4* lipase and overexpressed *DGA2* and *GPD1* genes under the control of the *pTEF* promoter. Deletion of lipase TGL4 and all six POX enzymes blocked the β-oxidation cycle and degradation of TAGs. Overexpression of the *DGA2* and *GPD1* genes leads to enhanced TAG production.

In opposition to *pTEF*, the *8UAS-pTEF* promoter is carrying eight upstream activating sequences enhancing its effectivity. In total, three strains were constructed, JMY7078 (*8UAS-pTEF-MsexD2*), JMY7080 (*8UAS-pTEF-MsexD3*), and JMY7084 (*8UAS-pTEF-MsexD2* and *8UAS-pTEF-MsexD3*).

Strains JMY7078, JMY7080, and JMY7084 were cultured in a MedA^+^ medium with glucose as the carbon source. There were almost no differences in biomass and lipid production among strains. All strains produced around 12 g/L of dry cell weight (DCW), of which 7.5 g/L consisted of lipid-free biomass and 4.5 g/L of lipids (Figure 4). Thus, the lipid content was slightly below 40% of DCW.

GC analysis was performed for the characterization and comparison of FA profiles of all three strains. The C18:1Δ^9^ acid was determined as the major FA (more than 45%) in the intracellular lipids in all strains (Figure 5). The use of the *8UAS-pTEF* promoter enhanced the production of the already detected *M. sexta* FAD metabolites and a trace amount of 2UFAs but did not contribute to the production of 3UFAs. Surprisingly, while in JMY6700, *M. sexta* FAD metabolites were detected, and analysis of the FAs profile showed that JMY7080 expressing MsexD3 desaturase under the control of *8UAS-pTEF* promoter did not produce any of these products. The reason for this result is that overexpression of other stronger lipid accumulation pathways overshadowed the new metabolic pathways, the result of which failed to show even though they were expressed under a stronger promoter. Both MsexD2 expressing strains (JMY7078 and JMY7084) contained C15:1Δ^11^, C16:1Δ^11^, C17:1Δ^11^, C18:1Δ^11^, C18:1Δ^13^, and C20:1Δ^11^ as monounsaturated FAs (Figure 6). JMY7084 contained additionally C14:1Δ^11^, C16:1Δ^13^, C16:2Δ^trans10,trans12^, and C16:2Δ^trans10,cis12^. While the production of Δ11 desaturated FAs was directly catalyzed by *M.sexta* FADs, Δ13 desaturated FAs originated via elongation. Despite the fact that metabolic modifications of JMY7084 also promoted the production of a trace amount of 2UFAs, no 3UFAs were detected by GC analysis. Only two strains were able to produce specific *M.sexta* FAD metabolites, especially C16:1Δ^11^ (Figure 6). The amount of C16:1Δ^11^ was highly similar in both strains (30.6 µg/mg DCW in JMY7078 vs. 32.1 µg/mg DCW in JMY7084). JMY7084 was chosen to be the most suitable for further experiments because as the double transformant with a complete metabolic pathway, it has a higher potential to achieve diverse FAs.

### 3.3. Supplementation of Media with Fatty Acid Methyl Esters

To boost the biosynthesis of *Msex*D2 and *Msex*D3 specific FAs, the JMY7084 strain, expressing *Msex*D2 + *Msex*D3 desaturases, and the medium was supplemented with biosynthetic FA precursors C16:0-Me, C16:1Δ^11^-Me, C16:2Δ^10,12^-Me dissolved in ethanol. The influence of FA additives was controlled by the cultivation of JMY7084 without supplements. Cells produced approximately the same amount of biomass in all media and accumulated similar amounts of lipids (Table 2). The effect of supplemented fatty acids on the total fatty acid profile and the production of FAs produced by *M. sexta* FADs is seen in Figure 7.

The addition of palmitic acid methyl ester to the medium did not increase the production of C16:1Δ^11^ (43.91 µg/mg DCW in JMY7084 vs. 43.02 µg/mg DCW in JMY7084 + C16:0-Me) and C16:2Δ^10,12^ (0.34 µg/mg DCW in JMY7084 vs. 0.35 µg/mg DCW in JMY7084 + C16:0-Me), nor increase C16:0 alone (79.85 µg/mg DCW in JMY7084 vs. 68.17 µg/mg DCW in JMY7084 + C16:0-Me).

Supplementation with the cis isomer C16:1Δ^11^ methyl ester increased the amount of C16:1Δ^11^ alone in the cells (43.91 µg/mg DCW in JMY7084 vs. 60.24 µg/mg DCW in JMY7084 + C16:1 Δ^11^-Me) and the amount of C16:2Δ^10,12^ (0.34 µg/mg DCW in JMY7084 vs. 0.48 µg/mg DCW in JMY7084 + C16:1 Δ^11^-Me), but the amount of C18:1Δ^13^ that arose in the elongation process increased markedly (0.68 µg/mg DCW in JMY7084 vs. 1.1 µg/mg DCW in JMY7084 + C16:10 Δ^11^-Me). The increase of C18:1Δ^13^indicates that *Y. lipolytica* naturally accumulates fatty acids with a chain length of 18 carbons. The fact that 18-carbon-long oleic acid is dominant in the fatty acid profile is already seen when looking at the FA profile of the wild-type strain. Thus, all potential new fatty acids with 16 carbons also have their elongated counterparts with 18 carbons chains.

The addition of C16:2Δ^10,12^-Me resulted in an increase of the FA (0.34 µg/mg DCW in JMY7084 vs. 0.76 µg/mg DCW in JMY7084 + C16:1 Δ^10,12^-Me) and an evident increase in elongated C18:2Δ^12,14^ (0.0 µg/mg DCW in JMY7084 vs. 0.74 µg/mg DCW in JMY7084 + C16:1 Δ^10,12^-Me), which confirms that *Y. lipolytica* accumulates mostly 18 carbons long fatty acids. Unfortunately, no supplements stimulated the synthesis of quantifiable amounts of 3UFAs.

With each addition of FAs, GCxGC-MS detected the formation and increase of new peaks. Using GCxGC-MS, we managed to identify 20 new FAs with two or three double bonds, some of which contained conjugated, isolated double bonds, or had a combination of both, but the content was too low for quantification of these new substances by GC-FID (Appendix A).

## 4. Discussion

The objective of this study was to metabolically engineer strains of the oleaginous yeast *Yarrowia lipolytica* to express desaturases native to *Manduca sexta* and thus produce specific FA derivatives, which consist mainly of C16:2Δ^trans10,cis12^ and C16:3Δ^trans10,trans12,cis14^, produced from C16:1Δ^cis11^ (Figure 8). The first strains of *Y. lipolytica* carrying the *M. sexta* genes were JMY6699 (*MsexD2*) and JMY6700 (*MsexD3*), both constructed out of the Po1d strain. TFA of JMY6699 contained 9.6% (17.8 µg/mg DCW) of C16:1Δ^cis11^, 2% (3.8 µg/mg DCW) of C18:1Δ^cis11^, and a trace amount of C17:1Δ^cis11^, while JMY6700 produced only traces of C16:1Δ^cis11^ as the new metabolite. Neither 2UFAs nor 3UFAs fatty acids were detected in our engineered *Y. lipolytica* strains. When *MsexD2* and *MsexD3* were expressed in *Saccharomyces cerevisiae*, MsexD2 desaturase showed both Δ11 desaturase and 10, 12—conjugase activities. On the other hand, the production of a mixture of C16:3Δ^trans10,trans12,trans14^ and C16:3Δ^trans10,trans12,cis14^ by MsexD3 was observed exclusively when C16:2Δ^trans10,trans12^ precursor was supplemented into the medium [11,26]. It can be seen from the results that non-oleaginous *S. cerevisiae* yielded better results than oleaginous *Y. lipolytica.* A possible explanation is that the very efficient catabolism of fatty acids in *Y. lipolytica* could eliminate the novel, unnatural fatty acids. Therefore, a different strategy was employed to enable the conjugated FAs production in *Y. lipolytica*, and desaturase genes were overexpressed in the JMY3820 strain [20], which was constructed previously and allowed the very efficient accumulation of triacylglycerols (TAG). Genes coding acyl-CoA: diacylglycerol acyltransferase *DGA2* and glycerol-3-phosphate dehydrogenase *GPD1* were overexpressed under the control of the *pTEF* promoter. To prevent TAG degradation and the degradation of fatty acids, the gene *TGL4* encoding lipase and all genes encoding acyl-CoA oxidases *POX1-6* were deleted. The efficient expression of *MsexD2* and *MsexD3* was driven by a strong constitutive *8UAS-pTEF* promoter [16]. *MsexD2* and *MsexD3* were inserted individually and in combination to mimic their natural activity in *M. sexta*. Enhanced production was observed in a strain expressing only *MsexD2* (JMY7078) and a strain expressing both *MsexD2* and *MsexD3* (JMY7084), showing us the importance of *Msex*D2 desaturase in a metabolic pathway of 3UFA creation. Both strains contained multiple monounsaturated FAs affected by the presence of the heterologously expressed desaturase (C15:1Δ^11^, C16:1Δ^11^, C17:1Δ^11^, C18:1Δ^11^, C18:1Δ^13^, and C20:1Δ^11^). JMY7084 additionally produced C14:1Δ^11^, C16:1Δ^13^, and trace amount of C16:2Δ^trans10,trans12^ and C16:2Δ^trans10,cis12^. The production of C16:1Δ^11^ reached values of 30.6 µg/mg DCW in JMY7078 vs. 32.1 µg/mg DCW in JMY7084, which corresponds to 7.9% of TFA in JMY7078 and 8.2% of TFA in JMY7084. Even though the neosynthesis conditions did stimulate the production of the higher amounts of the FA with one double bond and a trace amount of MsexD2 FA with two double bonds, the production of any C16 trienoic FA was not detected despite the coexpression of both desaturase in the same strain. To simulate the production of C16 trienoic FA, precursors (C16:0, C16:1Δ^cis11^, and C16:2Δ^trans10,cis12^) of 3UFAs were supplemented into the medium as was described by Buček et al. (2015) [11]. With the help of GCxGC-MS, we identified 20 new FAs, however most of them in trace amounts (Appendix A). In comparison, the *S. cerevisiae* activity of MsexD3 resulted in the production of monounsaturated FAs with a 14- or 16-carbon long chain. Production of C16:1Δ^cis11^ by MsexD3, a precursor of monounsaturated pheromones, was significantly lower than by MsexD2. In contrast to MsexD2, MsexD3 did not exhibit conjugase activity of C16:1Δ^cis11^ to C16:2Δ^trans10,trans12^ and C16:2Δ^trans10,cis12^. Heterologous expression of FAD, together with precursor supplementation, allowed *S. cerevisiae* to store only small amounts of major products (C16:1Δ^cis11^, C16:2Δ^trans10,cis12^, and C16:3Δ^trans10,trans12,cis14^) in the cells, and in addition, trace amounts of elongation-produced by-products (C16:1 and C16:2 FAs with Δcis13, or Δtrans13 double bond) were produced [11,26]. Very similar results were obtained with *Y. lipolytica*. Increasing the concentration of FAs, which are substrates for *Msex*D2 and *Msex*D3, caused an increase in both diene- and triene- FA production. In addition, desaturase precursors and their products became substrates for *Y. lipolytica* elongases, while undesirable by-products were formed, reducing the content of the desired diene and triene C16 FA. The issue of elongase interaction with novel FA metabolites was also addressed by Buček et al. 2015. However, the attempts to eliminate interfering yeast FA metabolites by expression of the *FAD* genes in the *S. cerevisiae* strain with deleted *ELO1* and *OLE1* genes, which cause a deficiency of fatty acyl desaturation and medium-chain fatty acyl elongation, led to the production of only trace amounts of novel FAs. Finally, both enzymes were characterized in *S. cerevisiae* W303, which has a single FAD with Δ9 desaturase activity and an active elongase system. However, the elongation in *Y. lipolytica* is more complicated. Rigouin et al. (2018) have described two elongases, *Yl*ELO1 and *Yl*ELO2, in *Y. lipolytica*. It was proved that elongase ELO1 serves to extend the chain from C14 to C16 and from C16 to C18, and elongase ELO2 serves to extend FAs from C16 to C18 and from C18 to longer chains. Since both enzymes are supposed to have the ability to elongate C16 to C18, the deletion of both would be necessary. However, it was shown that the deletion of *Yl*ELO2 seriously impaired the fitness of cells [27]. C18 FAs in *Y. lipolytica* are also produced by FAS (fatty acid synthase). It is possible to increase the production of C16 FAs with the modification of FAS, as described by Rigouin et al. (2017) [28]. However, the combination of FAS modification and ELO1 deletion caused cells to be unable to synthesize C18 FAs as necessary for membrane structure and survival without oleic acid supplementation into the medium.

It is evident that natural FAs accumulation has an important influence on the production of FAs by recombinant strains. For *Y. lipolytica*, it is natural for it to accumulate mainly 18-carbon-long FAs, with oleic acid as the main component of TFA, similar to *S. pombe*. While CLA and CLNA are both 18-carbon-long FAs derived from OA and LA as precursors, which are natural for *Y. lipolytica*, *M. sexta* pheromone precursors are 16-carbon-long FAs. Competition of native *Y. lipolytica* desaturation and elongation pathways and new heterologous pathways caused the formation of new unnatural FAs with unconventional double bond positions.

To conclude, our results prove that *Y. lipolytica* is capable of synthesizing C16 3UFAs, and this research is the first step on the long journey that the production of these substances represents. Despite these positive results, the production of shorter-chain conjugated FAs in *Y. lipolytica* will require further genetic optimizations of the genome and optimizations of the fermentation process. However, it will enable, for example, the production of precursors of some insect pheromones and other biotechnologically interesting derivatives.

## Figures and Tables

**Figure 1 jof-09-00114-f001:**
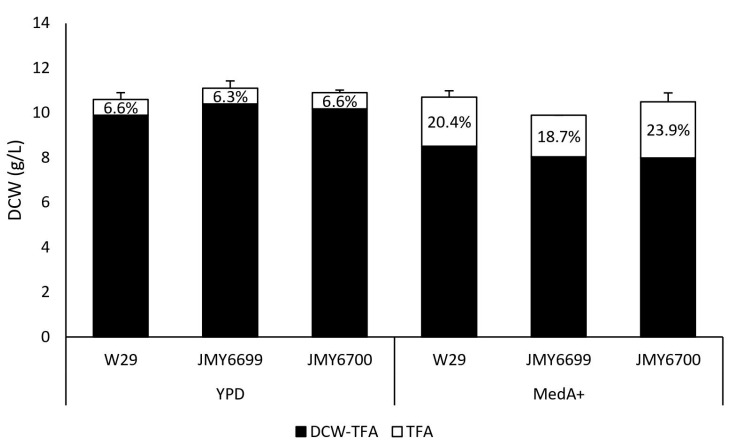
Biomass and lipid accumulation in W29 (control), JMY6699 (*pTEF-MsexD2*), JMY6700 (*pTEF-MsexD3*) cultured in two different media (YPD, 48 h; MedA+, 72 h) with glucose as the carbon source. Textured areas represent Lipid free biomass (g/L) and grey areas represent total fatty acids (g/L). Lipid accumulation (%) is expressed as a TFA to DCW ratio. Each value is an average of the values obtained from three independent experiments.

**Figure 2 jof-09-00114-f002:**
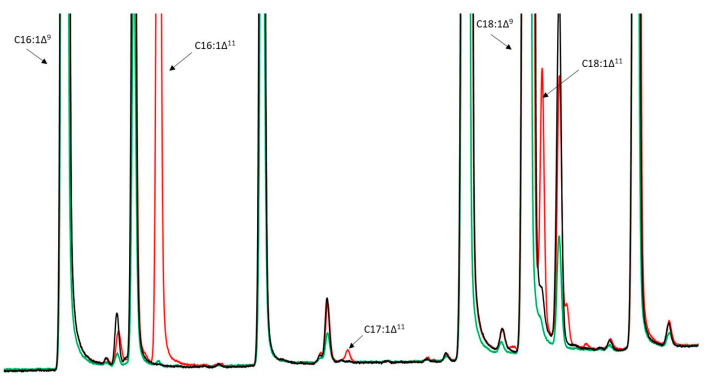
GC analysis of fatty acids in *Y. lipolytica* strains expressing *MsexD2* and *MsexD3*, grown on MedA^+^ medium for 72 h. W29—wild-type strain (control), JMY6699 expressing *MsexD2* and JMY6700 expressing *MsexD3*.

**Figure 3 jof-09-00114-f003:**
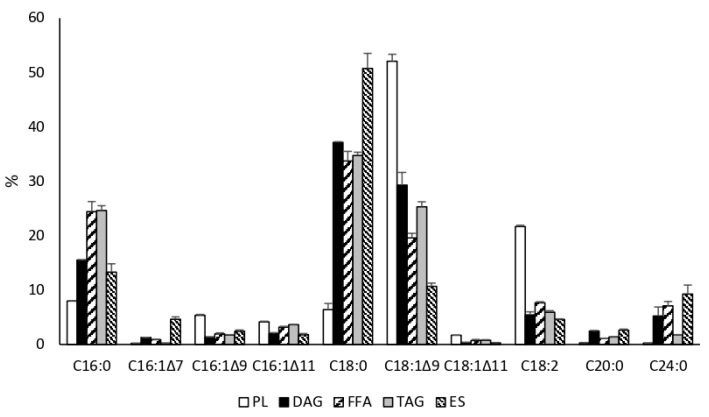
The FA composition of different lipid classes in JMY6699 (*pTEF-MsexD2*). PL—polar lipids, DAG—diacylglycerols, FFA—free fatty acids, TAG—triglycerides, ES—sterol esters.

**Figure 4 jof-09-00114-f004:**
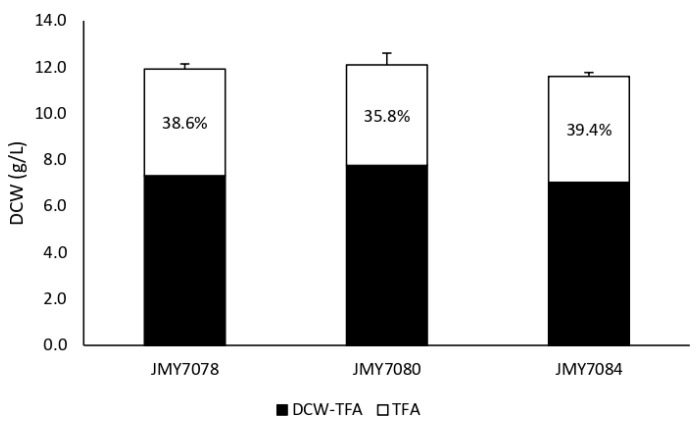
Biomass and lipid accumulation of JMY7078 (*8UAS-pTEF-MsexD2*), JMY7080 (*8UAS-pTEF-MsexD3*), JMY7084 (*8UAS-pTEF-MsexD2, 8UAS-pTEF-MsexD3*) cultured in MedA+ with glucose as the carbon source. Textured area represents Lipid free biomass (g/L) and grey area represents total fatty acids (g/L). Lipid accumulation (%) is expressed as a TFA to DCW ratio. Each value is an average of the values obtained from three independent experiments.

**Figure 5 jof-09-00114-f005:**
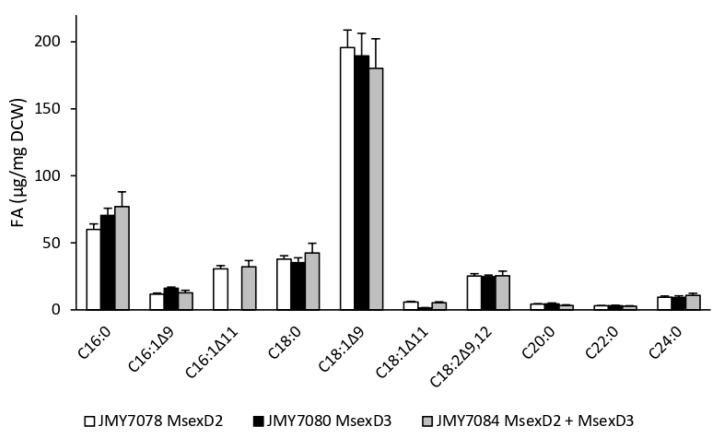
Profiles of the major FAs of the strains JMY7078 (*8UAS-pTEF-MsexD2*), JMY7080 (*8UAS-pTEF-MsexD3*), JMY7084 (*8UAS-pTEF-MsexD2, 8UAS-pTEF-MsexD3*) cultured in MedA+ with glucose as the carbon source. The values provided are an average of the values obtained in three parallel experiments.

**Figure 6 jof-09-00114-f006:**
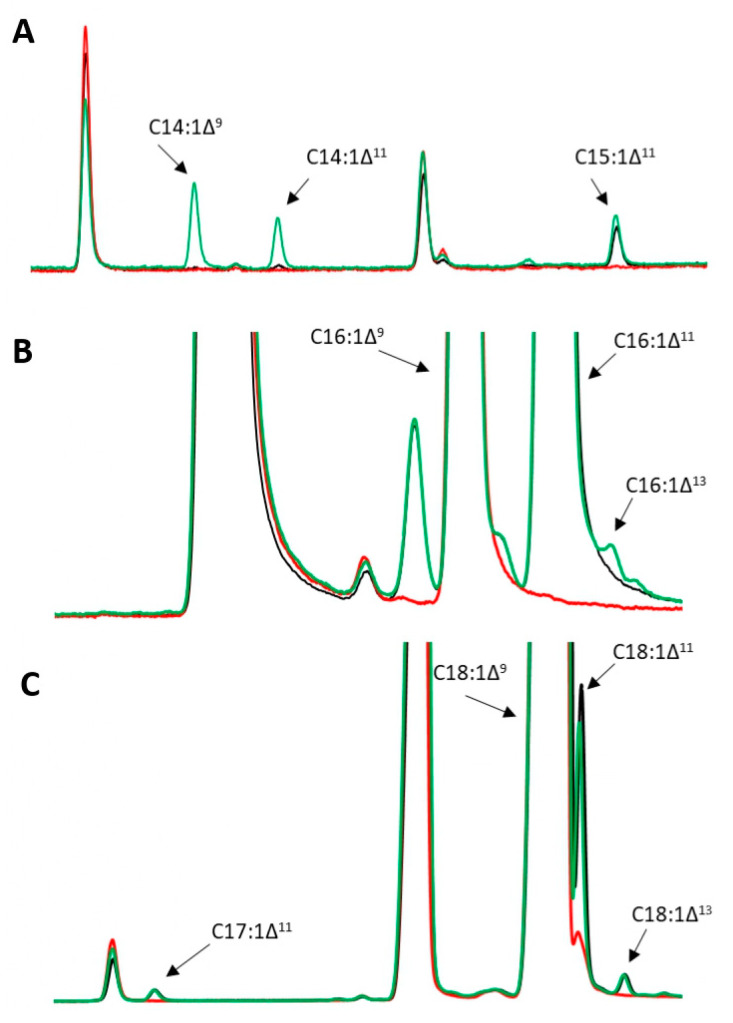
GC analysis of *M. sexta* FAD monounsaturated products, (**A**) 7 min–9.5 min; (**B**) 9.2 min–10 min; (**C**) 10 min–12.5 min. Marked new products: blue—JMY7078 (*Msex*D2), red—JMY7080 (*Msex*D3), and green—JMY7084 (*Msex*D2 + *Msex*D3). Strains were cultured in MedA+ with glucose as the carbon source.

**Figure 7 jof-09-00114-f007:**
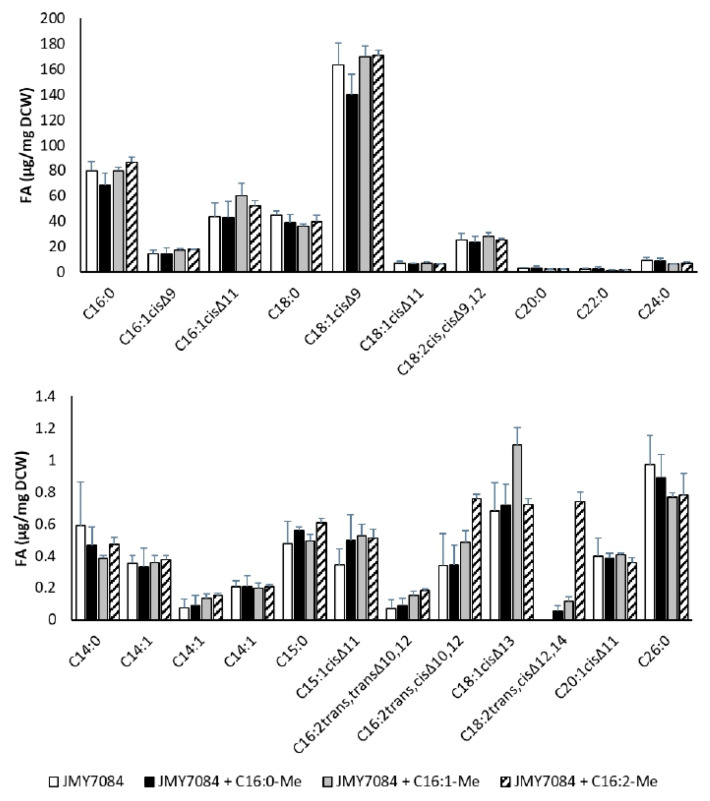
Fatty acid profiles of the strains JMY7084 (control without additives), JMY7084 + C16:0-Me (cultivation with C16:0-Me supplementation), JMY7084 + C16:1-Me (cultivation with C16:1Δ^cis11^-Me supplementation), and JMY7084 + C16:2-Me (cultivation with C16:2Δ^trans10,cis12^-Me supplementation) cultured in MedA+ with glucose as the carbon source. The values provided are an average of the values obtained in three parallel experiments.

**Figure 8 jof-09-00114-f008:**
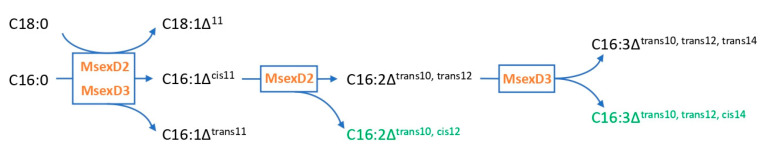
*Manduca sexta* metabolic pathway of pheromone precursors synthesis by MsexD2 and MsexD3 enzymes. Items highlighted orange are enzymes. Those highlighted green are main pheromone precursors.

**Table 1 jof-09-00114-t001:** Microorganisms and plasmids used in the study.

Strain (Host Strain)	Plasmid/Genotype	References
*Escherichia coli*		
JME1046	*JMP62-pTEF-URA3ex*	Lazar et al. 2013 [15]
JME2563	*JMP62-pTEF-LEU2ex*	Dulermo et al. 2017 [16]
JME2607	*JMP62-8UAS-pTEF-RedStae2-LEU2ex*	Dulermo et al. 2017 [16]
JME3048	*JMP62-8UAS-pTEF-URA3ex*	Dulermo et al. 2017 [16]
JME 4145	*JMP1046-MsexD2*	This work
JME 4147	*JMP1046-MsexD3*	This work
JME 4299	*JMP3048-MsexD2*	This work
JME 4301	*JMP2607-MsexD3*	This work
*Yarrowia lipolytica*		
W29	MATA, wild type	Barth and Gaillardin 1997 [17]
Po1d	MATA *leu2–270 ura3–302 xpr2–322 + pXPR2-SUC2*	Barth and Gaillardin 1997 [17]
JMY6699	Po1d, *pTEF-MsexD2-URA3ex, LEU2*	This work
JMY6700	Po1d, *pTEF-MsexD3-URA3ex, LEU2*	This work
JMY3501	W29 *ura3–302 leu2–270 xpr2–322 Δpox1–6 Δtgl4 + pXPR2-SUC2 + pTEF-DGA2-LEU2ex + pTEF-GPD1-URA3ex*	Lazar et al. 2014 [18]
JMY3820	W29 *ura3–302 leu2–270 xpr2–322 Δpox1–6 Δtgl4 + pXPR2-SUC2 + pTEF-DGA2 + pTEF-GPD1*	Lazar et al. 2014 [18]
JMY7078	JMY3820, *8UAS-pTEF-MsexD2-URA3ex, LEU2*	This work
JMY7080	JMY3820, *8UAS-pTEF-MsexD3-LEU2ex, URA3*	This work
JMY7084	JMY3820, *8UAS-pTEF-MsexD2-URA3ex, 8UAS-pTEF-MsexD3-LEU2ex*	This work

**Table 2 jof-09-00114-t002:** Biomass and lipid accumulation of JMY7084 (control without supplement), JMY7084 + C16:0-Me (cultivation with C16:0-Me supplementation), JMY7084 + C16:1-Me (cultivation with C16:1Δ^cis11^-Me supplementation), and JMY7084 + C16:2-Me (cultivation with C16:2Δ^trans10,cis12^-Me supplementation) cultured in MedA+ with glucose as the carbon source. DCW—dry cell weight, TFA—total fatty acids. Each value is an average of the values obtained from three independent experiments.

Strain + Addition	DCW (g/L)	TFA/DCW (%)
JMY7084	11.6 ± 0.6	40.1 ± 6.7
JMY7084 + C16:0-Me	12.4 ± 0.3	35.5 ± 6.4
JMY7084 + C16:1-Me	12.4 ± 0.5	41.6 ± 2.4
JMY7084 + C16:2-Me	12.5 ± 0.3	41.9 ± 0.6

## Data Availability

Not applicable.

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
