# Peer review of "Biosynthesis of Fatty Acid Derivatives by Recombinant Yarrowia lipolytica Containing MsexD2 and MsexD3 Desaturase Genes from Manduca sexta"

_jof, 2023, doi:10.3390/jof9010114_

Round 1
Reviewer 1 Report
Biosynthesis of fatty acid derivatives by recombinant Yarrowia lipolytica containing MsexD2 and MsexD3 desaturase genes from Manduca sexta
Thank you for your submission. This is a good proof of concept study. Please address the following comments/questions.
Ø Please provide reference for the following sentence: “modified microorganisms are much easier accepted by industry and society.”
Ø Please provide information on the C/N molar ratio of MedA+ medium used for the lipid production. Was that C/N=80?
Ø How did you determine the final concentration of FAME to provide/add?
Ø How much ethanol had to be transferred to the medium with the FAME, when (at which stage) did you add the FAME to the production culture?
Ø Are uncommon C16 long conjugated fatty acids with two and three conjugated double bonds considered as functional fatty acids with health benefits. If so, please provide information and corresponding reference on the clinically proved health benefits of the foregoing unsaturated fatty acids.
Ø It might be worth checking out this reference https://www.degruyter.com/document/doi/10.1515/znc-2017-0031/html
What type of vector in terms of copy number you have used? Or did you perform multiple copy integration? If so, how did you do the screening afterwards?
Can you elaborate on the substrate specificity of expressed desaturases. Is that accurate to say that expressed enzymes are active on the esterified fatty acids and not on free fatty acids?
Did the overexpression of desaturase have any effect on the lipid accumulation?
Line 386 – 391 Please make the sentence shorter.
Did you study the enzymatic activity of desaturases outside the cell? What is the advantage of whole cell bioconversion as compared to cell free reaction in the presence of purified or crude desaturases?
Author Response
Ø Please provide reference for the following sentence: “modified microorganisms are much easier accepted by industry and society.”
Corrected.
Ø Please provide information on the C/N molar ratio of MedA+ medium used for the lipid production. Was that C/N=80?
Yes C/N ratio in MedA+ medium was 80. It was also added into manuscript.
Ø How did you determine the final concentration of FAME to provide/add?
FAME concentration of 0,25mM was determined according to the protocol (Buček et al., 2015) that expressed same desaturases in S. cerevisiae.
Buček, A.; Matoušková, P.; Vogel, H.; Šebesta, P.; Jahn, U.; Weißflog, J.; Svatoš, A.; Pichová, I. Evolution of Moth Sex Pheromone Composition by a Single Amino Acid Substitution in a Fatty Acid Desaturase. Proc. Natl. Acad. Sci. 2015, 112, 12586–12591, doi:10.1073/pnas.1514566112
Ø How much ethanol had to be transferred to the medium with the FAME, when (at which stage) did you add the FAME to the production culture?
0.125mL of ethanol solution was transferred to the production medium before inoculation.
Ø Are uncommon C16 long conjugated fatty acids with two and three conjugated double bonds considered as functional fatty acids with health benefits. If so, please provide information and corresponding reference on the clinically proved health benefits of the foregoing unsaturated fatty acids.
We are not aware of any existing studies that have produced or studied the pharmaceutical properties of these new substances.
Ø It might be worth checking out this reference https://www.degruyter.com/document/doi/10.1515/znc-2017-0031/html
Thank you for your comment. We know this paper. In a fact, this review paper is a paper of our two co-authors (Pichova, Tupec) and we deeply discuss it.
What type of vector in terms of copy number you have used? Or did you perform multiple copy integration? If so, how did you do the screening afterwards?
A linear fragment bordered by retrotransposonal zeta sequences was used as a vector, thanks to which the DNA was integrated directly into the chromosome of Y. lipolytica.
Can you elaborate on the substrate specificity of expressed desaturases. Is that accurate to say that expressed enzymes are active on the esterified fatty acids and not on free fatty acids?
Substrates for M. sexta desaturases are fatty acyl-CoAs. In addition, the MsexD2 is bifunctional Z-D11 -desaturase, which exhibits Z11-desaturase and conjugase (1,4-dehydrogenase) activity and catalyzes the production of Z11-hexadecenoate (Z11–16) and Z10E12- and E10E12-hexadecadienoates (Z10E12–16) via 1,4-desaturation of the Z11–16 substrate. MsexD3 does not not exhibit conjugase activity and it catalyzes production of E10,E12,E14-16:3 and E10,E12,Z14-16:3 from E10, E12-16:2. Additional specific products of MsexD3 are Z11-16:1, Z11-14:1, and E11-14:1. Z11-16:1,
Did the overexpression of desaturase have any effect on the lipid accumulation?
The desaturases were overexpressed in the strain with different genetical background. Significant change in lipid production occurred due to extensive interventions in lipid metabolism of strain JMY3501 compared to a wild type strain. Therefore, the effect of increased expression of desaturases on lipid production cannot be determined.
Line 386 – 391 Please make the sentence shorter.
Corrected.
Did you study the enzymatic activity of desaturases outside the cell? What is the advantage of whole cell bioconversion as compared to cell free reaction in the presence of purified or crude desaturases?
MsexD2 and MsexD3 are membrane proteins and their activities have never been characterized outside the cells. Their specificities correspond to the composition of pheromones synthetized in M. sexta. Specificity of MsexD3 was confirmed by topically applied metabolic probes in the form of FAs and FAMEs to female M. sexta pheromone glands using deuterium-labeled E10,E12-16,16,16-2H3-16:2 methyl ester. In vitro desaturase activity and specificity have been performed only with soluble plant desaturases.
Reviewer 2 Report
Please see the attachment.

Author Response
Point 1: page 2, line 53 – include a transcript for FAs
Corrected and included in the text.
Point 2: page 2, lines 61-62 - include a transcript for 2UFA and 3UFA
Corrected and included in the text.
Point 3: page 2, line 80 – correct to Escherichia coli
Corrected.
Point 4: page 2, line 81 – correct to E. coli
Corrected.
Point 5: page 2, line 96-98 – check the salt formulas
Corrected.
Reviewer 3 Report
The review at hand deals with the posibility pathways biosynthesis of fatty acid derivatives by recombinant Yarrowia lipolytica containing MsexD2 and MsexD3 desaturase genes from tobacco horn-worm (Manduca sexta). The topic is interesting, and it can attract a wide range of readers. The authors have experience in this field and the overall organization of the manuscript seems appropriate. The article has some shortcomings/inaccuracies that should be clarified/corrected.
Introduction
Please describe the potential benefits of biosynthesis C16 long conjugated fatty acids with two and three conjugated double bonds. What are the health-promoting properties of C16 conjugateds?
Results:
The figure 1 shows the average results from 3 replicates? please add standard deviations.
The results of authors prove that Y. lipolytica is capable to synthetize C16 3UFAs, but their trace amounts, however, raise the question of reasonableness in the face of the costs of their synthesis.
Author Response
Introduction
Please describe the potential benefits of biosynthesis C16 long conjugated fatty acids with two and three conjugated double bonds. What are the health-promoting properties of C16 conjugateds?
We are not aware of any existing studies that have produced or studied the pharmaceutical properties of these new substances. We only asume that these new fatty acids could have some interesting health benefits, because of the health capabilities conjugated C18 fatty acids possess. Future higher production would allow us to test pharmaceutical properties of conjugated C16:3 FAs.
Results:
The figure 1 shows the average results from 3 replicates? please add standard deviations.
The standard deviations are already in the figure.
The results of authors prove that Y. lipolytica is capable to synthetize C16 3UFAs, but their trace amounts, however, raise the question of reasonableness in the face of the costs of their synthesis.
This is only the first proof of concept study. Future optimizations of Y. lipolytica strain and process of fermentation to improve the production is certainly needed.